# Health, lifestyle and sociodemographic characteristics are associated with Brazilian dietary patterns: Brazilian National Health Survey

**Jonas Eduardo Monteiro dos Santos**[1¤]*, **Sandra Patricia Crispim**[2], **Jack Murphy**[3], **Marianna de Camargo Cancela**[1,4]

1 Division of Population Research, Brazilian National Cancer Institute (INCA), Rio de Janeiro, Brazil, 2 Department of Nutrition, Federal University of Paraná (UFPR), Curitiba, Brazil, 3 Department of Epidemiology, University of North Carolina at Chapel Hill, Chapel Hill, North Carolina, United States of America, 4 Division of Surveillance and Situation Analysis, Brazilian National Cancer Institute, Rio de Janeiro, Brazil

¤ Current address: National School of Public Health, Oswaldo Cruz Foundation (FIOCRUZ), Rio de Janeiro, Brazil

* monteirojonaseduardo@gmail.com

## Abstract

This study aimed to identify Brazilian dietary patterns and their associations with health, lifestyle and sociodemographic characteristics. Data from the Brazilian National Health Survey conducted in 2013 were used. A questionnaire was applied containing 22 items related to dietary consumption. Dietary patterns were determined through factor analysis (FA). Poisson regression models, with robust variance, were used to identify associations between dietary patterns and independents variables. Statistical significance was defined as p-value<0.05. Data were analysed for 60,202 adults (estimated population size: 146,308,458). FA identified three dietary patterns: healthy, protein, and western. The younger age group (18–24 years) had a lower adherence to the healthy pattern (PR:0.53; 95% CI:0.49–0.58) and greater adherence to the protein (PR:1.52; 95%CI:1.42–1.62) and western (PR:1.80; 95%CI:1.68–1.93) patterns compared to the elderly (≥60 years). Women had a greater association with the healthy pattern (PR:1.32; 95%CI:1.28–1.38) and lower association with the protein pattern (PR:0.80; 95%CI:0.77–0.82) compared to men. Illiterate participants showed lower adherence to the healthy (PR:0.58; 95%CI:0.53–0.63) and western (PR:0.54; 95%CI:0.48–0.62) patterns compared to those with higher educational levels. Smokers had lower adherence to the healthy (PR:0.76; 95%CI:0.71–0.81) and higher adherence to the protein (PR:1.14; 95%CI:1.11–1.19) patterns compared to non-smokers. Participants with poor/very poor self-rated health status had a lower adherence to the healthy (PR:0.79; 95%CI:0.73–0.86) and western (PR:0.81; 95%CI:0.73–0.89) patterns compared to those in a very good/good self-rated health status. Multimorbidity was positively associated with the healthy pattern (PR:1.18; 95%CI:1.11–1.26) and inversely associated with the protein pattern (PR:0.88; 95%CI:0.80–0.96) compared to participants without

**Data Availability Statement:** The data underlying the results presented in the study are available from https://www.ibge.gov.br/estatisticas/sociais/saude/9160-pesquisa-nacional-de-saude.html?=&t=o-que-e.

**Funding:** The author(s) received no specific funding for this work.

**Competing interests:** The authors have declared that no competing interests exist.

comorbidities. We suggest that strategies to promote healthy eating should consider health, lifestyle and sociodemographic characteristics in the Brazilian population.

## Introduction

Dietary patterns have changed throughout human history. Food industry modernization contributes to continuous and abundant access to energy-dense food rich in lipids, sugar, and additives [1]. On the other hand, fruit, vegetable, and fish production and consumption are dependent on season, region, climate, and food system sophistication [1]. Changes in dietary patterns associated with government initiatives have contributed to reducing disease prevalence related to nutritional deficiencies [1]. However, this has created space for non-communicable diseases (NCDs) [1]. These changes in diet and disease patterns are known as the nutritional and epidemiological transition [1]. In Brazil, NCDs are responsible for 70% of deaths. One third of these deaths occur in people under the age of 60 [2]. Vulnerable people—especially those with lower income and education—are the most affected by NCDs. Social inequality is thus a key factor in the NCD burden [2].

Studies associating dietary patterns, sociodemographic characteristics and lifestyle factors are of particular interest in the nutritional epidemiology field [3]. Most studies consider food intake or nutrient effects in isolation, disregarding the complex interactions between them [3,4]. However, a typical diet is composed of foods combined in meals and snacks with a high degree of interactions between nutritional compounds [4]. Thus, dietary patterns have been proposed as an alternative to fill this methodological gap [3].

Most studies that evaluated dietary patterns have focused on countries in North America, Europe, and Asia [5]. Few Brazilian studies are available, and the ones that are focus on specific areas or specific population groups [6–14]. Moreover, the most recent studies evaluating national dietary patterns are relatively old, having used data from 2002/2003 [10,15].

Therefore, studies using recent data representing the Brazilian population are needed to monitor the ongoing changes in Brazilian dietary patterns. New evidences will contribute to the development and improvement of public policies aiming to reduce NCDs. This study aims to identify the main Brazilian dietary patterns and associated health, lifestyle and sociodemographic characteristics.

## Materials and methods

### Design and study population

This cross-sectional study used data from the Brazilian National Health Survey—BNHS (in Portuguese: Pesquisa Nacional de Saúde—PNS), 2013 [16]. BNHS is a five-year household survey with a complex sample design, representative of the Brazilian population in macro-regions, states, urban, rural, and metropolitan areas [17]. The Brazilian Ministry of Health and the Brazilian Institute of Geography and Statistics (IBGE) developed and carried out the BNHS. It is the largest national survey conducted to date on Brazilian public health [18]. The sample design used conglomerates in three levels: primary units of sampling (first stage), household (second stage), and selection of the household adult resident (third stage) [19]. Indigenous villages, barracks, military bases, housing camps, boats, penitentiaries, penal colonies, prisons, asylums, orphanages, convents, and hospitals were not included in the sampling design [17].

The questionnaire consisted of three parts: (i) housing and neighbourhood; (ii) household, and (iii) individual head-of-household (applied to a randomly selected resident older than 18)

[17]. The National Commission of Ethics in Research for Human Beings approved the survey under number 328.159. All participants gave free and informed written consent [19]. More details about BNHS methods are available in a previous publication [19].

## Variables

The following independent variables were selected for analysis: age, sex, colour/race, marital status, education, area of residence, macro-regions, socioeconomic status, physical activity, tobacco use, alcohol intake, self-rated health status and multimorbidity.

Age was categorized into four groups: 18–24 years; 25–39 years; 40–59 years; 60 years and older. Skin colour was self-declared as white, black, east asians, brown or indigenous and then categorised into "white or east asians" and "other". In Brazil, whites and east asians have similar socioeconomic characteristics [20]. Education was categorised as illiterate, elementary school, high school and university.

Socioeconomic status was defined based on availability of household items and the provision of home services and correspondent scores. The method used is described in greater detail in a previous publication [21]. The cut-off criteria for each socioeconomic class were as follows: Class A (45–100 points); B (29–44 points); C (17–28 points); D and E (0–16 points) [21].

Physical activity level included activities practised during leisure time, work, domestic activities, and commuting. This variable was categorised as sufficient ($\geq$ 150 minutes per week), insufficient ($>0$ and $<150$ minutes per week), and sedentary (no physical activity) [22].

Tobacco use was categorised into non-smokers, ex-smokers, or smokers, and was based on self-reported consumption. Alcohol intake was categorised as abstainer, moderate, or binge drinker according to the Brazilian Ministry of Health [18,23].

The presence of two or more NCDs in the same person was defined as multimorbidity [24]. The following diseases were considered chronic conditions: systemic arterial hypertension, diabetes mellitus, hypercholesterolaemia, cardiovascular disease, asthma, arthritis or rheumatism, vertebral problems, work-related musculoskeletal disorder, depression, mental illness, lung disease, cancer (all types) and chronic kidney disease. The variable was classified into four categories (none or 1, 2, 3, and 4 conditions or more). Physical and mental health were also self-assessed according to the question: in general, how do you assess your health? Possible answers were very good/good, regular, bad/very bad.

## Statistical analysis

The food consumption questionnaire was composed of 22 food items. Factor Analysis (FA) was performed to identify dietary patterns. FA allows the reduction of a large number of variables to a few factors that are not correlated with each other, explaining patterns underlying the original data.

The Bartlett test and Kaiser–Mayer–Olkin (KMO) coefficient were applied to verify the method applicability. KMO$\geq$0.6 was considered acceptable in the partial correlation between diet variables [25,26]. For the Bartlett test, we assumed a Type I error rate of 5% [27,28].

Each dietary pattern was composed of food items with factor loads $\leq$-0.35 and $\geq$0.35. The total variances explained by each factor were also considered to determine the number of factors to be retained. Cattell's scree plots and Eigenvalue $>1$ were also considered to select the patterns [28]. Orthogonal varimax transformation was applied to facilitate the interpretation of factor results [28]. Internal consistency between diet items and their respective factors was evaluated using Cronbach's alpha test [29]. Participants received a factorial score for each identified pattern. Dietary patterns were categorized into quartiles (Q1-Q4). We assumed the first quartile as the reference quartile and compared it with the others. Thereby, we sought to

observe associations with independents variables according by higher or lower adherence to each dietary pattern.

We applied a Poisson regression model with robust variance to estimate the association between dietary patterns, health and sociodemographic characteristics. Robust Poisson regression models allow correction for overestimation of associations when outcomes are not rare (>10% prevalence). Associations were presented as Prevalence Ratios (PR) with 95% confidence intervals (95% CI). Taking into account the cross-sectional desing of the study, we considered one as a constant time at risk to estimate the prevalence ratios [30].

The independents variables were age, sex, colour/race, marital status, educational level, area of residence (urban/rural), physical activity, smoking, alcohol intake, self-rated health, multimorbidity and socioeconomic status. Wald tests were used to evaluate the significance of variables in the bivariate analyses; values with p≤0.20 were retained for the multivariate model. Only variables with significant associations (Wald test p<0.05) to the dietary patterns were kept in the final model. All analyses were stratified by Brazilian macro-regions and performed with Stata software, version 14 [31].

## Results

### Health, sociodemographic and lifestyle characteristics

Data from 60,202 participants were analysed (participation rate 93.6%), representing 146,308,458 Brazilian adults according to the complex sample design.

Table 1 shows the study population characteristics. The average age was 41.3 years (SD: ± 16.6) and 53.0% were women. Thirty four percent of the population attended at least high school and 37.9% primary school only. Approximately 86% of the population lived in urban areas, and 43.8% lived in the southeast region. More than half (54%) reported practicing enough physical activity. The prevalence of non–smokers was 67.8% and smokers 14.7%. The prevalence of binge drinking was 13.6%. Approximately 66% of the sample self-reported very good or good health. Those reporting none or one NCD were 76.4%.

### Factor analysis

Table 2 presents the results of the FA, which identified three main dietary patterns: healthy, protein, and western. Fig 1 shows the scree plot revealed three main dietary patterns with eigenvalues greater than one (see Table 2 for details). The healthy pattern was composed of raw salads, vegetables, cooked vegetables, fruits, and fresh fruit juice. The protein pattern was composed of beans, red meat, fish, and chicken. The western pattern was composed of processed and ultra-processed foods: snacks, pizzas, sweets, and soft drinks. The three main patterns explained 40% of diet variability. The overall KMO was 0.61. The p-value for the Bartlett test was 0.001. The highest p-value for the Cronbach Alpha test was 0.55. Milk was not included in any of the dietary patterns due to a low factorial load.

### Association of dietary patterns with independents variables

Table 3 shows the associations between the first quartile (Q1—reference quartile) and the last quartile (Q4) of each pattern in the Brazilian population. Associations between the first, second and third quartiles are presented in the supplement as are all results related to the stratified analysis by macro-regions.

Nationally, adherence to the healthy pattern was less common among those aged 18–24 (PR: 0.53; 95%CI: 0.49–0.58) than among those 60 and older. Associations between age and healthy pattern were similar in all Brazilian macro-regions. A dose-response relationship

**Table 1. Estimated population size (N = 146,308,458), prevalence (% per column), and confidence intervals (95% CI) of helath, lifestyle and sociodemographic characteristics in Brazilian adults.**

| Variables | N | % (95%CI) |
|---|---|---|
| **Age group (years)** | | |
| Mean | | 41.3 |
| SD | | 16.6 |
| 18–24 | 23,306,033 | 16.0 (15.0–16.5) |
| 25–39 | 46,494,908 | 31.7 (31.1–32.5) |
| 40–59 | 50,099,686 | 34.3 (33.6–35.0) |
| 60+ | 26,407,831 | 18.0 (14.5–18.7) |
| **Gender** | | |
| Men | 68,916,470 | 47.0 (46.3–47.9) |
| Women | 77,391,988 | 53.0 (52.0–53.6) |
| **Skin Colour/Race** | | |
| White/East Asian | 70,813,082 | 48.5 (47.6–49.3) |
| Other [a] | 75,495,376 | 51.5 (50.8–52.4) |
| **Marital status** | | |
| Married | 89,537,328 | 61.2 (60.5–61.2) |
| Other [b] | 56,771,130 | 38.8 (38.0–39.5) |
| **Education** | | |
| University | 26,958,232 | 19.3 (18.5–20.2) |
| High School | 50,173,018 | 34.3 (33.6–35.0) |
| Elementary School | 54,004,400 | 37.9 (37.0–38.7) |
| Illiterate | 11,948,795 | 8.6 (8.2–9.0) |
| **Area of residence** | | |
| Urban | 126,132,422 | 86.2 (85.7–86.6) |
| Rural | 20,176,036 | 13.8 (13.3–14.2) |
| **Economic status** | | |
| A-B | 36,633,476 | 25.0 (24.5–25.6) |
| C | 57,463,271 | 39.3 (38.6–40.0) |
| D-E | 52,211,711 | 35.7 (35.1–36.3) |
| **Physical Activity** | | |
| Sufficient | 78,933,914 | 54.0 (53.1–54.7) |
| Insufficient | 26,540,646 | 18.0 (17.5–18.7) |
| None | 40,833,898 | 28.0 (27.1–28.6) |
| **Smoking** | | |
| Never | 99,248,243 | 67.8 (67.1–68.5) |
| Ex-smokers | 25,540,840 | 17.5 (16.8–18.0) |
| Current | 21,519,375 | 14.7 (14.2–15.2) |
| **Alcohol intake** | | |
| Abstainer | 87,183,278 | 59.6 (58.7–60.4) |
| Moderate | 39,152,545 | 26.8 (26.0–27.5) |
| Binge drinker | 19,972,635 | 13.6 (13.1–14.2) |
| **Self-Rated Health** | | |
| Very good/Good | 96,748,777 | 66.1 (65.4–66.8) |
| Fair | 41,039,237 | 28.0 (27.4–28.7) |
| Poor/Very poor | 8,520,444 | 5.9 (5.5–6.1) |
| **Multimorbidity** | | |
| 0 or 1 | 111,769,410 | 76.4 (75.7–77.0) |

*(Continued)*

**Table 1.** (Continued)

| Variables | N | % (95%CI) |
|---|---|---|
| 2 | 18,245,024 | 12.5 (12.0–13.0) |
| 3 | 8,901,041 | 6.9 (5.7–6.5) |
| 4+ | 7,392,983 | 5.2 (4.7–5.4) |

N, Absolute frequency of estimated population. CI, confidence interval. SD, standard deviation.

[a] Black(a), brown(a), indigenous.

[b] single, divorced, separated, widowed.

between age and healthy pattern was observed in all Brazilian macro-regions and at the national level. The healthy pattern was significantly more common among women compared to men in all regions and in Brazil as a whole.

The healthy pattern was significantly less frequent among illiterate participants compared to those with a university education in Brazil as a whole and all macro-regions. Likewise, the healthy pattern was significantly more common among urban residents than rural residents in the North (PR: 0.49; 95% CI: 0.37–0.66) and Northeast (PR: 0.71; 95% CI: 0.61–0.83) macro-regions. On the other hand, rural inhabitants of the South region were more likely to adopt the healthy pattern (PR: 1.15; 95% CI: 1.04–1.27) than those living in urban areas in the same region (Supplement material).

Women were more likely to adopt the healthy pattern than men (PR: 1.21; 95% CI: 1.17–1.26). Adherence to the healthy pattern was more common among married than unmarried individuals. Adherence to the healthy pattern was less common among current smokers, binge drinkers, and those with self-rated health as poor/very poor compared to non-smokers, abstainers and people with very good/good self-rated health, respectively. Individuals with

**Table 2. Brazilian dietary patterns components identified by Factor Analysis (FA).**

| Dietary Patterns | Food Items/Food Group | Factor Load | KMO | Eigenvalue | Variance (%) [a] | CronbachAlpha test |
|---|---|---|---|---|---|---|
| Healthy | Lettuce and tomato salad or other vegetable or raw legumes | 0.75 | 0.60 | 1.91 | 15.60 | 0.55 |
| | Cooked vegetables and legumes [b] | 0.73 | 0.60 | | | |
| | Natural fruit juice | 0.42 | 0.63 | | | |
| | Fruits | 0.59 | 0.72 | | | |
| Protein | Beans | 0.54 | 0.56 | 1.64 | 12.80 | 0.39 |
| | Red meat | 0.68 | 0.58 | | | |
| | Fish | -0.62 | 0.61 | | | |
| | Poultry | -0.38 | 0.52 | | | |
| Western | Soft drinks | 0.60 | 0.60 | 1.26 | 11.80 | 0.41 |
| | Sweets [c] | 0.65 | 0.60 | | | |
| | Sandwiches, snacks or pizza | 0.70 | 0.59 | | | |

KMO, Kaiser-Meyer-Olkin.

Total variance:40.20. Total KMO: 0.61.

[a] The variance percentage explained by each pattern.

[b] cabbage, carrot, chayote, eggplant, pumpkin. Not considered: Potato, cassava, and yams–foods rich in starch.

[c] cakes, pies, chocolates, candies, biscuits, or sweet biscuits.

Bartlett's sphericity test p-value<0.001.

Factor loads ≤-0.35 and ≥0.35.

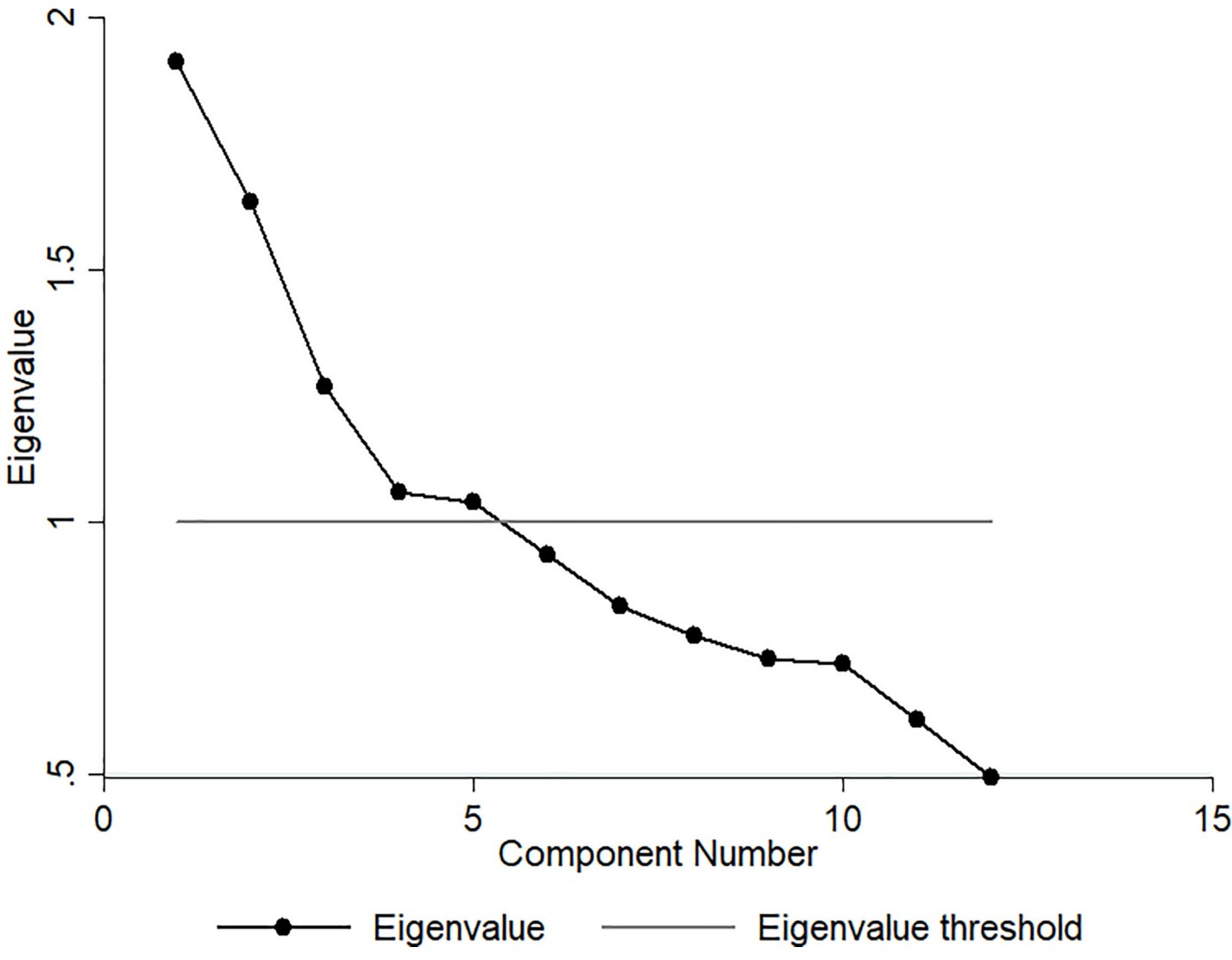

**Fig 1. Cattell's scree plot.**

four or more NCD's were more likely to adopt the healthy pattern than those with none or one NCD.

Younger participants (18–24 years) were more likely to adopt the protein pattern (PR: 1.52; 95% CI: 1.42–1.62) than the elderly (≥60 years), with significant associations nationally and in all Brazilian macro-regions. In addition, women were significantly less likely to adopt the protein pattern than men. Married individuals were more likely to adopt the protein pattern than their unmarried counterparts. The protein pattern was more common among illiterate participants (RP: 1.60; 95% CI: 1.47–1.73) compared to those with a university degree. Current smokers and binge drinkers were more likely to adopt the protein pattern than non-smokers and abstainers. The adherence to the protein pattern was less common among those with four or more NCDs compared to those with none or one NCD. Nationally, associations between skin color, economic status, physical activity, self-rated health and the protein pattern were not significant.

**Table 3. Associations between health, lifestyle, sociodemographic characteristics and countrywide Brazilian dietary patterns.** Comparison between first and fourth quartiles.

| DIETARY PATTERNS | HEALTHY | | PROTEIN | | WESTERN | |
|---|---|---|---|---|---|---|
| Prevalence Ratio | Crude (95%CI) | Adjusted (95%CI) | Crude (95%CI) | Adjusted (95%CI) | Crude (95%CI) | Adjusted (95%CI) |
| Sample Size (n) | 30,101 | | 30,101 | | 30,101 | |
| Estimated Population Size (N) | 72,190,359 | | 72,135,995 | | 75,207,779 | |
| **Age groups (years)** | | | | | | |
| 60+ | 1.00 | 1.00 | 1.00 | 1.00 | 1.00 | 1.00 |
| 18–24 | 0.59(0.55–0.64) | 0.53(0.49–0.58) | 1.38(1.30–1.47) | 1.52(1.42–1.62) | 2.37(2.21–2.54) | 1.80(1.68–1.93) |
| 25–39 | 0.75(0.72–0.79) | 0.68(0.64–0.71) | 1.33(1.26–1.41) | 1.40(1.33–1.48) | 1.90(1.77–2.05) | 1.49(1.38–1.60) |
| 40–59 | 0.87(0.83–0.91) | 0.81(0.78–0.85) | 1.22(1.16–1.29) | 1.20(1.14–1.26) | 1.34(1.25–1.45) | 1.16(1.08–1.24) |
| p-value | <0.005 | <0.005 | <0.005 | <0.005 | <0.005 | <0.005 |
| **Sex** | | | | | | |
| Male | 1.00 | 1.00 | 1.00 | 1.00 | 1.00 | - |
| Female | 1.32(1.28–1.38) | 1.21(1.17–1.26) | 0.73(0.70–0.75) | 0.80(0.77–0.82) | 1.02(0.98–1.06) | - |
| p-value | <0.005 | <0.005 | <0.005 | <0.005 | 0.261 | - |
| **Skin Color/Race** | | | | | | |
| White/Yellow | 1.00 | 1.00 | 1.00 | - | 1.00 | 1.00 |
| Other[a] | 0.72(0.70–0.75) | 0.92(0.89–0.96) | 1.00(0.96–1.03) | - | 0.72(0.69–0.75) | 0.90(0.86–0.93) |
| p-value | <0.005 | <0.005 | 0.882 | - | <0.005 | <0.005 |
| **Marital status** | | | | | | |
| Other[b] | 1.00 | 1.00 | 1.00 | 1.00 | 1.00 | - |
| Married | 1.10(1.06–1.15) | 1.08(1.04–1.12) | 1.09(1.06–1.13) | 1.08(1.05–1.12) | 0.86(0.83–0.89) | - |
| p-value | <0.005 | <0.005 | <0.005 | <0.005 | <0.005 | - |
| **Education** | | | | | | |
| College | 1.00 | 1.00 | 1.00 | 1.00 | 1.00 | 1.00 |
| High School | 0.78(0.75–0.82) | 0.90(0.86–0.94) | 1.37(1.29–1.45) | 1.34(1.27–1.42) | 0.88(0.85–0.91) | 0.92(0.89–0.96) |
| Elementary School | 0.69(0.65–0.72) | 0.76(0.72–0.79) | 1.46(1.38–1.55) | 1.54(1.46–1.64) | 0.54(0.52–0.57) | 0.73(0.69–0.76) |
| Illiterate | 0.49(0.44–0.54) | 0.58(0.53–0.63) | 1.20(1.11–1.30) | 1.60(1.47–1.73) | 0.30(0.27–0.34) | 0.54(0.48–0.61) |
| p-value | <0.005 | <0.005 | <0.005 | <0.005 | <0.005 | <0.005 |
| **Area of residence** | | | | | | |
| Urban area | 1.00 | 1.00 | 1.00 | 1.00 | 1.00 | 1.00 |
| Rural area | 0.60(0.56–0.65) | 0.83(0.78–0.89) | 1.10(1.05–1.15) | 1.17(1.12–1.22) | 0.47(0.43–0.52) | 0.64(0.59–0.69) |
| p-value | <0.005 | <0.005 | <0.005 | <0.005 | <0.005 | <0.005 |
| **Economic Status** | | | | | | |
| A-B | 1.00 | 1.00 | 1.00 | - | 1.00 | - |
| C | 0.84(0.80–0.88) | 1.01(0.96–1.05) | 1.04(1.00–1.09) | - | 0.82(0.78–0.85) | - |
| D-E | 0.67(0.64–0.70) | 0.92(0.88–0.97) | 0.98(0.93–1.02) | - | 0.66(0.63–0.70) | - |
| p-value | <0.005 | <0.005 | <0.005 | - | <0.005 | - |
| **Physical Activity** | | | | | | |
| Sufficient | 1.00 | 1.00 | 1.00 | - | 1.00 | - |
| Insufficient | 0.89(0.84–0.94) | 0.86(0.82–0.90) | 1.01(0.97–1.06) | - | 0.95(0.91–1.00) | - |
| None | 0.86(0.82–0.90) | 0.83(0.80–0.87) | 0.99(0.95–1.03) | - | 0.86(0.82–0.90) | - |
| p-value | <0.005 | <0.005 | 0.598 | - | <0.005 | - |
| **Smoking** | | | | | | |
| Never | 1.00 | 1.00 | 1.00 | 1.00 | 1.00 | - |
| Ex-smoker | 0.92(0.88–0.97) | 0.91(0.87–0.96) | 1.05(1.01–1.10) | 1.04(1.00–1.09) | 0.77(0.73–0.81) | - |
| Current | 0.66(0.62–0.71) | 0.76(0.71–0.81) | 1.28(1.24–1.33) | 1.15(1.11–1.19) | 0.86(0.82–0.91) | - |
| p-value | <0.005 | <0.005 | <0.005 | <0.005 | <0.005 | - |

*(Continued)*

**Table 3.** (Continued)

| DIETARY PATTERNS | HEALTHY | | PROTEIN | | WESTERN | |
|---|---|---|---|---|---|---|
| Prevalence Ratio | Crude (95%CI) | Adjusted (95%CI) | Crude (95%CI) | Adjusted (95%CI) | Crude (95%CI) | Adjusted (95%CI) |
| **Alcohol intake** | | | | | | |
| Abstainer | 1.00 | 1.00 | 1.00 | 1.00 | 1.00 | 1.00 |
| Moderate | 0.96(0.92–1.00) | 0.95(0.92–0.99) | 1.13(1.09–1.18) | 1.02(0.98–1.06) | 1.31(1.26–1.36) | 1.09(1.06–1.14) |
| Binge drinker | 0.72(0.67–0.77) | 0.84(0.79–0.90) | 1.29(1.24–1.34) | 1.09(1.05–1.14) | 1.31(1.25–1.38) | 1.10(1.06–1.15) |
| p-value | <0.005 | <0.005 | <0.005 | <0.005 | <0.005 | <0.005 |
| **Self-Rated Health** | | | | | | |
| Very good/Good | 1.00 | 1.00 | 1.00 | - | 1.00 | 1.00 |
| Fair | 0.85(0.81–0.89) | 0.87(0.84–0.91) | 0.92(0.88–0.95) | - | 0.66(0.63–0.69) | 0.90(0.86–0.95) |
| Poor/Very poor | 0.74(0.67–0.80) | 0.79(0.73–0.86) | 0.86(0.80–0.93) | - | 0.48(0.43–0.54) | 0.81(0.73–0.89) |
| p-value | <0.005 | <0.005 | <0.005 | - | <0.005 | <0.005 |
| **Multimorbidity** | | | | | | |
| 0 or 1 | 1.00 | 1.00 | 1.00 | 1.00 | 1.00 | - |
| 2 | 1.15(1.08–1.21) | 1.03(0.98–1.08) | 0.83(0.77–0.90) | 0.96(0.91–1.01) | 0.79(0.75–0.84) | - |
| 3 | 1.31(1.24–1.39) | 1.13(1.07–1.20) | 0.77(0.71–0.85) | 0.93(0.87–1.00) | 0.72(0.66–0.79) | - |
| 4+ | 1.34(1.27–1.41) | 1.18(1.11–1.26) | 0.77(0.71–0.85) | 0.88(0.80–0.96) | 0.65(0.59–0.73) | - |
| p-value | <0.005 | <0.005 | <0.005 | 0.012 | <0.005 | - |

Wald Test: Variables not statistically significant in the model.

[a] Black(a), brown(a), indigenous.

[b] single, divorced, separated, widowed.

Younger participants also showed higher adherence to the western pattern (PR: 1.80; 95% CI: 1.68–1.93) compared to the elderly. Adherence to the western pattern was less frequent among black, brown and indigenous participants compared to white and East Asian participants together. Overall, the western pattern was significantly less common among illiterate participants (PR: 0.54; 95% CI: 0.48–0.61) compared to those holding a university degree. In the North, rural residence was inversely associated with the western pattern compared to those living in the urban area (PR: 0.29; 95% CI: 0.21–0.40). Rural dwellers in other macro-regions also presented a weaker association with the western pattern compared to urban area residents (Supplementary Information). The western pattern was more frequent among binge drinkers than abstainers (PR: 1.10; 95% CI: 1.06–1.15). On the other hand, the western pattern was less common among those with poor/very poor self-rated health compared to those with very good/good self-rated health. Nationally, associations between sex, marital status, economic status, physical activity, smokers, multimorbidity and the western pattern were not statistically significant.

## Discussion

Our study identified three main dietary patterns–healthy, protein and western–which explained 40% of the diet variability. Adherence to the healthy pattern was less common among younger participants (18–24 years old), black, brown and indigenous participants, illiterate people, people living in rural areas, physically inactive individuals, current smokers, binge drinkers and those with lower socioeconomic status and self-rated poor/very poor health. The protein pattern was more common among people who were younger, married, male, illiterate, living in rural areas, current smokers, binge drinkers and those with none or one NCD. Finally, the western pattern was more likely to be adopted by younger individuals,

binge drinkers, those holding a university degree and those living in urban areas. To our knowledge, this is the first study to analyse dietary patterns using a nationally representative sample of Brazilian population.

The healthy pattern presented high factor loads for the healthy eating markers. Our findings corroborate previous studies that found similar dietary patterns [32], with high factor loads for vegetables, fruits and fresh fruit juice, poultry, low-fat cheeses, roots, tubers and fish [8,9,12,15]. The healthy pattern has also been described in other studies as vegetables [33] or prudent [9,34–44], with similar composition.

Previous studies evaluating dietary patterns in Brazilian subpopulations also identified the western pattern. High factor loads were observed for foods such as butter, margarine, added sugar, bread, pasta, fats, dairy products, sauces, pizza, processed meat, soft drinks, canned vegetables, sweets and desserts [9,13,14]. The western pattern is also described as unhealthy [34,36–38,40,41,43–51] or modern [36], presenting a high factor load for red and processed meat, eggs, refined grains, cookies, snacks, pizza, French fries and hamburgers.

Healthy and western patterns have been verified in many dietaries patterns studies and describe the extremes of a group diet. Between these patterns are intermediate dietaries patterns which characterize the differences in eating habits in different groups. Our study identified the protein pattern as the intermediate pattern. Intermediate patterns have been described in previous literature as mixed pattern presenting a high factor load for cereals, eggs, soft drinks, coffee, juice, fruit, vegetables, nuts, dairy products, butter, margarine, meat, fish, shrimp, sweets and alcohol [6]. The snacks pattern is composed of butter, cheese, meat, pork, beef, processed meats, sandwiches, eggs, dairy products, sweets, and desserts. Finally, the dual pattern [8,9,52] consists of a high factor load of dairy products, fresh fruit, tomato, vegetables, juice, fruits, green vegetables, bananas, sweets, desserts, soft drinks, processed meats, fast food, margarine, and cookies [10]. The traditional Brazilian pattern is an intermediate pattern [6–15,25,52–57] characterised by rice and beans consumption [15]. This pattern was not identified in this study because the BNHS did not evaluate rice consumption. The BNHS considered only healthy eating markers, and the rice consumption assessment was not prioritized, even though beans are usually consumed with rice in Brazil. Other intermediate country-specific patterns described in the literature are traditional Polish [42], Kimchi rice [58], Vegetarian [59], Tex–Mex [48,49], Dim Sum [60], Mediterranean [38,45], and Mchicha [61] patterns.

## Sociodemographic characteristics and dietary patterns

Previous studies identified associations between dietary patterns, educational level, and income in the Brazilian population [15]. However, other sociodemographic characteristics were not considered [62]. We found that age, sex, educational level, skin colour, marital status and socioeconomic status were associated with dietary patterns.

We observed dose-response effects in the associations between age groups and dietary patterns. These dose-response effects suggest age is an important factor in food choice. Previous studies have shown that the elderly prefer healthy diets [63–66]. On the other hand, younger people prefer soft drinks and fast food [4,65,67,68]. Several factors could explain healthier eating habits in older individuals. They have more time to prepare their meals and more knowledge about healthy eating habits than young people. Moreover, they are generally more concerned about diet as part of NCD prevention and control. The high NCD prevalence observed among the elderly leads them to adopt a healthier lifestyle, including diet improvement [69,70]. Adequate diet practices require knowledge, planning time for groceries, and cooking. For young people, these aspects can be considered barriers to adopting healthy eating habits. Therefore, they tend to choose more pragmatic alternatives: eating away from home,

usually high-energy density foods [15]. A study of 34,000 Brazilians reported that about 40% of participants consumed food outside home. Beetwen teenagers, fifty-one percent reported consumed food outside home. The most frequently consumed food group was the high-energy content and low nutritional quality one: alcoholic beverages, fried foods, pizza, soft drinks, and sandwiches [71]. Lack of time to prepare food, lack of access to healthy food in schools and universities, financial instability, poor culinary skills, little knowledge about food preparation, lack of cooking equipment, and easy access to processed products were indicated as the main reasons for eating unhealthily [72]. In this scenario, the provision of healthy food in school and university environments could be a strategy to promote healthy eating habits among young people.

Women presented higher adherence to the healthy pattern compared to men in all Brazilian macro-regions (S1–S5 Tables). On the other hand, women's adherence to the protein pattern was lower compared to men. Only in the Southeast of the country, the western pattern was more common among women compared to men (S1 Table). Culturally, women are more concerned about health and body shape, which also reflects on food choices. Our findings corroborate previous studies where women showed greater adherence to healthy patterns compared to men [4,63,64,68].

In the Southeast and the Northeast, married people had higher adherence to the healthy pattern compared to unmarried people (S1 and S4 Tables in Suplementary Information). We verified higher adhesion to the protein pattern in married people living in Southeast, South, Midwest, and North compared to unmarried people (S1, S2, S3 and S5 Tables in Suplementary Information). Previous findings have pointed to inconsistent associations between marital status and dietary patterns [65,73]. However, married life seems to have a positive effect on changing eating habits over time [73]. A study conducted in an urban population in Lithuania revealed that married individuals were more likely to follow a diet rich in fresh and cooked vegetables, fruits, eggs, tomatoes, and meats than single individuals [66]. Married individuals tend to prepare and consume healthier foods, while single individuals tend to choose fast food options.

People with black/brown skin colour and indigenous identify lower adherence to the healthy and western patterns. On the other hand, greater adherence to the protein pattern was seen among black, brown, and indigenous participants from the Southeast and South compared to whites and East Asians (S1 and S2 Tables). Previous studies reported differences in dietary patterns between different ethnic groups [64,74]. Ethnicity is related to social inequities and income distribution in Brazil and other developing countries. Indigenous and black/and brown people generally have less access to regular education, and have lower incomes [75–77]. These factors also affect their food choices. A study in the United States found that, among white people, educational level was strongly associated with healthier patterns; however, this association was weak among black people [78].

Adherence to the healthy and western patterns was less common among the lower education group. Associations were stronger between the protein pattern and low education groups. Dietary patterns may differ between educational levels [65,66] because people with higher education tend to adopt healthier eating habits [66,68,79,80]. Our results differ, in part, from previous findings: healthy and western patterns were more common among those with higher educational levels. In this study, educational level was used as a proxy for income. People with higher educational levels tend to have higher purchasing power, which may be associated with healthier food choices. Furthermore, people with higher education are more able to understand the importance of healthy eating habits [65].

Adherence to the healthy pattern was less frequent among people with lower economic status; this group was more likely to adopt the protein and western patterns when compared to

those with a higher economic status. Foods rich in saturated fat and sugar are cheaper than healthy and organic foods [15,65,66,74]. The adherence to the protein pattern among those with lower socioeconomic status could be explained by some factors: (i) among foods with high factor load in this pattern, beans are the protein source most affordable in Brazil; (ii) beans are part of a typical Brazilian meal consumed by every social classes; (iii) indigenous people cultivate and consume different species of beans in the North and Northeast regions of Brazil; (iv) the cultural high consumption of barbecue in the Midwest and South regions of Brazil; (v) fishing is common among indigenous people.

## Health, lifestyle characteristics and dietary patterns

Our study also identified statistically significant associations between the three dietary patterns, health and lifestyle characteristics in the Brazilian population. People with healthy eating habits tend to adopt a healthy lifestyle. Adherence to the healthy pattern was more common among physically active individuals, abstainers and non-smokers. Many researchers have shown that people who practice regular physical activity and do not smoke are more likely to consume fruits, fishes, vegetables, legumes and less red and processed meats [67,80–83].

On the other hand, binge drinkers and smokers were more likely to adopt the protein and western patterns. The consumption of alcoholic beverages can modulate appetite and food choices [84]. The diet among alcohol consumers tends to be lower in carbohydrates, fiber, vitamins, minerals, fruits, vegetables and dairy products [84–86] and high in animal products, oils, fatty acids, bread and breakfast cereals [85]. The consumption of unhealthy food is also frequent among smokers. Nicotine can modify smokers' sense of taste. With that, they prefer foods high in sugar and fat and processed food become more palatable [4,68,80,87].

Adherence to the healthy pattern was less frequent among individuals who reported poor/very poor self-rated heath, which corroborates a previous study [66]. Adherence to the western pattern was more frequent among those with very good/good self-rated health status in the Southeast and Midwest regions. These contradictory findings could be explained by some factors: (i) people with very good/good self-rated health are less concerned about healthy eating habits; (ii) people with NCDs tend to adopt healthier eating habits. Nationally, people with four or more NCDs were more likely to adopt the healthy pattern. Otherwise, the protein pattern was less frequent among those with four or more NCDs. We highlight as a limitation that our study is a cross-sectional study design in which reverse causality may be present. In addition, self-rated health status is a subjective variable.

We highlight as a strength of this study the fact that the BNHS is the most comprehensive, and representative health survey conducted in Brazil. Our study was able to perform dietary patterns analysis with a representative sample of the Brazilian population. In addition, we applied robust statistical methods to explore the relationship between the main dietary patterns, health, lifestyle and sociodemographic characteristics. Most previous Brazilian studies that derived dietary patterns only used samples from large urban centres [6–9,11,52,88]. This survey provided us with data from rural areas and other small towns, which enriched our findings. Lastly, FA evaluates the diet globally, considering all aspects of diet complexity.

Our study has some limitations. First, the available data in BNHS may not have captured all dietary patterns of the studied population [5]. This limitation is inherent to the instrument applied to collect food consumption data: BNHS applied a screen questionnaire of diet with only 22 food items that did not capture all foods consumed in Brazil. Second, associations between dietary patterns and independents variables presented in this study should be interpreted with caution. FA is not a mutually exclusive technique, and the same individual may have a high factor load for more than one pattern.

We identified three main dietary patterns in the Brazilian population: healthy, protein and western. We concluded that people with a healthy lifestyle were more likely to adopt healthy pattern. On the other hand, the western pattern was more common among those with an unhealthy lifestyle—smokers, binge drinkers and the physically inactive. Based on our findings, we suggest that strategies to promote healthy eating habits should consider these aspects related to health, lifestyle and sociodemographic characteristics.

## Supporting information

**S1 Table. Associations between dietary patterns, lifestyle, health and sociodemographic characteristics in the Southeast Region of Brazil.** Comparison between quartile 1 and quartile 4 for each dietary pattern.
(PDF)

**S2 Table. Associations between dietary patterns, lifestyle, health and sociodemographic characteristics in the South Region of Brazil.** Comparison between quartile 1 and quartile 4 for each dietary pattern.
(PDF)

**S3 Table. Associations between dietary patterns, lifestyle, health and sociodemographic characteristics in the Midwest Region of Brazil.** Comparison between quartile 1 and quartile 4 for each dietary pattern.
(PDF)

**S4 Table. Associations between dietary patterns, lifestyle, health and sociodemographic characteristics in the Northeast Region of Brazil.** Comparison between quartile 1 and quartile 4 for each dietary pattern.
(PDF)

**S5 Table. Associations between dietary patterns, lifestyle, health and sociodemographic characteristics in the North Region of Brazil.** Comparison between quartile 1 and quartile 4 for each dietary pattern.
(PDF)

**S6 Table. Associations between dietary patterns, lifestyle, health and sociodemographic characteristics in Brazil.** Comparison between quartile 1 and quartile 2 for each dietary pattern.
(PDF)

**S7 Table. Associations between dietary patterns, lifestyle, health and sociodemographic characteristics in the Southeast Region of Brazil.** Comparison between quartile 1 and quartile 2 for each dietary pattern.
(PDF)

**S8 Table. Associations between dietary patterns, lifestyle, health and sociodemographic characteristics in the South Region of Brazil.** Comparison between quartile 1 and quartile 2 for each dietary pattern.
(PDF)

**S9 Table. Associations between dietary patterns, lifestyle, health and sociodemographic characteristics in the Midwest Region of Brazil.** Comparison between quartile 1 and quartile 2 for each dietary pattern.
(PDF)

**S10 Table. Associations between dietary patterns, lifestyle, health and sociodemographic characteristics in the Northeast Region of Brazil.** Comparison between quartile 1 and quartile 2 for each dietary pattern.
(PDF)

**S11 Table. Associations between dietary patterns, lifestyle, health and sociodemographic characteristics in the North Region of Brazil.** Comparison between quartile 1 and quartile 2 for each dietary pattern.
(PDF)

**S12 Table. Associations between dietary patterns, lifestyle, health and sociodemographic characteristics in Brazil.** Comparison between quartile 1 and quartile 3 for each dietary pattern.
(PDF)

**S13 Table. Associations between dietary patterns, lifestyle, health and sociodemographic characteristics in the Southeast Region of Brazil.** Comparison between quartile 1 and quartile 3 for each dietary pattern.
(PDF)

**S14 Table. Associations between dietary patterns, lifestyle, health and sociodemographic characteristics in the South Region of Brazil.** Comparison between quartile 1 and quartile 3 for each dietary pattern.
(PDF)

**S15 Table. Associations between dietary patterns, lifestyle, health and sociodemographic characteristics in the Midwest Region of Brazil.** Comparison between quartile 1 and quartile 3 for each dietary pattern.
(PDF)

**S16 Table. Associations between dietary patterns, lifestyle, health and sociodemographic characteristics in the Northeast Region of Brazil.** Comparison between quartile 1 and quartile 3 for each dietary pattern.
(PDF)

**S17 Table. Associations between dietary patterns, lifestyle, health and sociodemographic characteristics in the North Region of Brazil.** Comparison between quartile 1 and quartile 3 for each dietary pattern.
(PDF)

## Author Contributions

**Conceptualization:** Jonas Eduardo Monteiro dos Santos, Sandra Patricia Crispim, Marianna de Camargo Cancela.

**Data curation:** Jonas Eduardo Monteiro dos Santos, Jack Murphy, Marianna de Camargo Cancela.

**Formal analysis:** Jonas Eduardo Monteiro dos Santos, Marianna de Camargo Cancela.

**Investigation:** Jonas Eduardo Monteiro dos Santos, Sandra Patricia Crispim, Marianna de Camargo Cancela.

**Methodology:** Jonas Eduardo Monteiro dos Santos, Sandra Patricia Crispim, Marianna de Camargo Cancela.

**Project administration:** Jonas Eduardo Monteiro dos Santos.

**Supervision:** Jonas Eduardo Monteiro dos Santos, Sandra Patricia Crispim, Marianna de Camargo Cancela.

**Validation:** Jonas Eduardo Monteiro dos Santos, Sandra Patricia Crispim, Marianna de Camargo Cancela.

**Visualization:** Jonas Eduardo Monteiro dos Santos, Sandra Patricia Crispim, Marianna de Camargo Cancela.

**Writing – original draft:** Jonas Eduardo Monteiro dos Santos, Sandra Patricia Crispim, Marianna de Camargo Cancela.

**Writing – review & editing:** Jonas Eduardo Monteiro dos Santos, Sandra Patricia Crispim, Jack Murphy, Marianna de Camargo Cancela.

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
