## [Decision Letter · Decision Letter 0]

21 Dec 2020

PONE-D-20-35214

Age, sex and sociodemographic differences are associated with Brazilian dietary pattern: National Health Survey

PLOS ONE

Dear Dr. dos Santos,

Thank you for submitting your manuscript to PLOS ONE. After careful consideration, we feel that it has merit but does not fully meet PLOS ONE’s publication criteria as it currently stands. Therefore, we invite you to submit a revised version of the manuscript that addresses the points raised during the review process.

We look forward to receiving your revised manuscript.

Kind regards,

Michele Drehmer, Ph.D

Academic Editor

PLOS ONE

Journal Requirements:

2. In statistical methods, please refer to any post-hoc corrections to correct for multiple comparisons during your statistical analyses. If these were not performed please justify the reasons. Please refer to our statistical reporting guidelines for assistance (https://journals.plos.org/plosone/s/submission-guidelines.#loc-statistical-reporting).

3. In your multivariable analyses, please state whether you accounted for clustering by locality. For example, did you consider using multilevel models?

4.We suggest you thoroughly copyedit your manuscript for language usage, spelling, and grammar. If you do not know anyone who can help you do this, you may wish to consider employing a professional scientific editing service.  

Reviewers' comments:

Reviewer's Responses to Questions

**Comments to the Author**

1. Is the manuscript technically sound, and do the data support the conclusions?

Reviewer #1: Partly

Reviewer #2: Yes

2. Has the statistical analysis been performed appropriately and rigorously? 

Reviewer #1: Yes

Reviewer #2: Yes

3. Have the authors made all data underlying the findings in their manuscript fully available?

Reviewer #1: Yes

Reviewer #2: Yes

4. Is the manuscript presented in an intelligible fashion and written in standard English?

Reviewer #1: Yes

Reviewer #2: Yes

5. Review Comments to the Author

Reviewer #1: The manuscript addresses a topic of potential interest of PLOS ONE readers. The objective was 'to identify Brazilian dietary patterns and their associations with sociodemographic characteristics'. The authors conclude ‘their results suggest that strategies to promote healthy eating habits should focus on the younger population'. The rationale/concept of the manuscript is interesting. The article provides a unique opportunity to explore the dietary patterns of the Brazilian population. However, I suggest a revision to clarify some basic concepts related to the analysis. I have following major and minor comments:

Major comments

1. Components was extracted using the Factor Analysis, not Principal Components Analysis instead. They are similar but not quite the same. Could you clarify this point to avoid any confusion?

2. These patterns represent a hypothetical composition, such as ‘healthy diets’ or ‘unhealthy'. Those concepts (health vs unhealthy) are not new and does not reflect the diversity of the ‘real diet’. Furthermore, the idea of the protein pattern is a new approach, the article is well constructed with an expressive Brazilian sample that will be helpful for future studies.

3. I suggest Cattell’s scree test, to visualize the proportion of variance explained by each component/factor (eigenvalues)

4. There are seemingly contradictory pieces of information such: (lines 299-306). 'People with black and brown skin colour and indigenous heritage presented lower adherence to the healthy and western patterns (...) greater adherence to the protein pattern'; 'Indigenous, black, and brown people generally have less access to regular education, have lower incomes and perform low-paid activities. These factors also affect their food choices'. However, the protein pattern is composed of beans, red meat, fish, and chicken. Can people from low-income levels buy protein patterns? Lines 318-320 contradicted the information above, about skin color/income and diet intake: 'positive association between the highest educational level and the western pattern, which could be explained by purchasing power'. The discussion about income/skin color/education was a bit confusing for me.

Minor comments:

1. lines127. ‘The Bartlett test and Kaiser–Mayer–Olkin (KMO) coefficient were applied to verify the method applicability’... The KMO is relevant in the context of FA, but not PCA, which uses the "Kaiser rule".

2. lines 128-129. 'KMO≥0.6 was considered appropriate in the partial correlation between diet variables’. Values of 0.6 are considered not best choices, would be at least >0.7 to be considerable as intermediate values according to Kaiser (1975)

3. line 180. ‘The highest p-value for the Cronbach Alpha test was 0.55’ Cronbach Alpha test is lower than references recommendations, indicates as limit cut above 0.70 as acceptable, however ≥0.80 would better. (Griethuijsen et al., 2014, Cortina, J. M., 1993).

4. line 180-181. ‘Milk was not included in any of the dietary patterns due to a low factorial load’. I believe that information must be explored in more depth.

5. line 230. 'which explained 40% of the diet variability'. Is this percentage considered enough for the total variance explanation?

6. line 265. ‘Several factors could explain healthier eating habits in older individuals.’ There is only one reference to explain the whole background. The authors should rewrite with more information about these hypotheses.

Reviewer #2: This study examined the association between sociodemographic characteristics and the main Brazilian dietary patterns. This manuscript addresses an interesting nutrition topic using data of the Brazilian population. The study has its strengths and, overall, the article is well written, but I have some major comments.

1. Title: Aren't age and sex sociodemographic characteristics? The use of the term "sociodemographic characteristics" is not enough in the title?

2. Title: “Brazilian dietary pattern” or “Brazilian dietary patterns”?

3. Introduction: There are many sentences in the first paragraph of "Introduction" text without references. This needs to be revised. Example (Page 2-line 47): “Changes in dietary patterns associated with government initiatives have contributed to reducing disease prevalence related to nutritional deficiencies.”

4. Design and Study Population: Is this a cross-sectional study? Why is this not described in the study design?

5. Variables (Page 4-line 93): “The selected variables were age, sex, colour/race, marital status, education, area of residence, macro-region, socioeconomic status, physical activity, tobacco use, alcohol intake, self-related health status and multimorbidity.” This section of the manuscript describes behavioral, health and multimorbidity variables. However, only sociodemographic variables were explored in the study. What is the main reason for adopting this procedure? Exploring behavioral, health and multimorbidity characteristics with dietary patterns would make the article more robust and interesting.

6. Statistical Analysis: The principal component analysis (PCA) appears to have been well conducted. The food consumption questionnaire was composed of 22 food items. Is this number of food items considered adequate? This could be highlighted or otherwise weighted among the study's limitations.

7. Statistical Analysis (Page 5-line 138): “Robust Poisson regression models allow correction for overestimation of associations when outcomes are not rare (>10% prevalence)(28).” I believe that there is a better reference for this information, considering cross-sectional studies. [Barros AJ, Hirakata VN. Alternatives for logistic regression in cross-sectional studies: an empirical comparison of models that directly estimate the prevalence ratio. BMC Med Res Methodol. 2003;3(1):21. DOI: 10.1186/1471-2288-3-21.]

8. Statistical Analysis (Page 5-line 141): “Associations were presented as Incidence Rate Ratios (IRR) with 95% confidence intervals (95% CI).” Is this the adequate measure of effect for the data under analysis? Is there any reason for not using the Prevalence Ratio?

9. Statistical Analysis: The description of the multivariate analysis model is confusing and needs to be revised. For example: Have demographic variables been adjusted to each other? Were behavioral, health and morbidity variables considered only as possible confounders? Why these variables were not explored as independent variables? If these characteristics are not explored as independent variables, I believe that it would be more appropriate to remove them from the analysis of the present study.

10. Results (Table 1): The data interpretation using the estimated sample size becomes confused. Is this form of presentation necessary?

11. Results (Page 10-line 194): “Table 3 shows the associations between the first quartile (Q1 - reference quartile) and the last quartile (Q4) of each pattern.” This procedure was not described and justified in the Methods section of the manuscript. Why was not the first quartile (Q1) compared to the sum of the remaining three (Q2 + Q3 + Q4)?

12. Results (Page 10-line 198): “In stratified analysis by macro-region, age showed a dose-response effect.” This sentence is not clear. Was this finding only for the regions and not for the national data (Brazil)?

13. Results: Table 3 is too long. I believe that this should be reduced. As most of the findings are similar across regions, Table 3 should only include national results (Brazil) and possible differences between regions should be described only in the text of the results.

14. Results: Table 3. The comparison groups (categories) of dietary patterns should be indicated in the title or heading of the Table.

15. Results: Table 3: The use of the following terms should be reviewed: “Crude” or Unadjusted? -- “(CI 95%)” or 95%CI? -- “<0.005” or <0.001?

16. Results: Table 3 (footnote): What mean "mutually adjusted models"?

17. Results: The ‘Results section’ should be thoroughly revised. Why the findings regarding ‘marital status’ and ‘skin color’ were not highlighted in the text, for example?

18. Discussion: First paragraph of the Discussion: why the main findings for the associations between sociodemographic factors and dietary patterns were not pointed out in the text?

19. Discussion (page 19-line 252): “The traditional Brazilian pattern is an intermediate pattern characterised by rice and beans consumption. This pattern was not identified in this study because the survey did not evaluate rice consumption.” Considering its regular consumption by the Brazilian population, what was the reason for this food item has not been evaluated?

20. Discussion (page 21-line 303): “Ethnicity is related to social inequalities and income distribution in Brazil and other developing countries.” This sentence needs reference (s).

21. Conclusion: The conclusion (last paragraph of the discussion) could be more explored and expanded. This could contemplate a synthesis of all sociodemographic aspects that showed an association in the study, in addition to age.

6. PLOS authors have the option to publish the peer review history of their article (what does this mean?). If published, this will include your full peer review and any attached files.

Reviewer #1: No

Reviewer #2: No

---

## [Author Response · Author response to Decision Letter 0]

27 Jan 2021

Reviewer #1: The manuscript addresses a topic of potential interest of PLOS ONE readers. The objective was 'to identify Brazilian dietary patterns and their associations with sociodemographic characteristics'. The authors conclude ‘their results suggest that strategies to promote healthy eating habits should focus on the younger population'. The rationale/concept of the manuscript is interesting. The article provides a unique opportunity to explore the dietary patterns of the Brazilian population. However, I suggest a revision to clarify some basic concepts related to the analysis. I have following major and minor comments:

Major comments

1. Components was extracted using the Factor Analysis, not Principal Components Analysis instead. They are similar but not quite the same. Could you clarify this point to avoid any confusion?

We apologize for the mistake, we actually performed Factor Analysis and this was duly correct into the text; line 132.

2. These patterns represent a hypothetical composition, such as ‘healthy diets’ or ‘unhealthy'. Those concepts (health vs unhealthy) are not new and does not reflect the diversity of the ‘real diet’. Furthermore, the idea of the protein pattern is a new approach, the article is well constructed with an expressive Brazilian sample that will be helpful for future studies.

We acknowledge reviewer’s comments, and we agree that it does not reflect the real diet; however, this is one of the limitations of nutritional epidemiology studies.

3. I suggest Cattell’s scree test, to visualize the proportion of variance explained by each component/factor (eigenvalues)

We included the scree plot in the manuscript, along with Table 2, which explains the partial variance of each dietary pattern and the total variance of the three patterns combined.

4. There are seemingly contradictory pieces of information such: (lines 299-306). 'People with black and brown skin colour and indigenous heritage presented lower adherence to the healthy and western patterns (...) greater adherence to the protein pattern'; 'Indigenous, black, and brown people generally have less access to regular education, have lower incomes and perform low-paid activities. These factors also affect their food choices'. However, the protein pattern is composed of beans, red meat, fish, and chicken. Can people from low-income levels buy protein patterns? Lines 318-320 contradicted the information above, about skin color/income and diet intake: 'positive association between the highest educational level and the western pattern, which could be explained by purchasing power'. The discussion about income/skin color/education was a bit confusing for me.

We apologize for the confusion and acknowledge the comment. The unclear topic was discussed in more detail in the manuscript; lines 372-381. We highlight that some specific characteristics in the Brazilian population and culture could explain these apparent contradictions: (i) among foods with high factor load in the protein pattern, beans are the protein source most affordable in Brazil. Rice and beans are components of a typical Brazilian meal consumed by every social class; (ii) indigenous cultivate and consume different species of beans in North and Northeast regions of Brazil; (iv) the cultural high consumption of barbecue in the Midwest and South regions of Brazil; (v) fishing is common among indigenous.

Minor comments:

1. lines127. ‘The Bartlett test and Kaiser–Mayer–Olkin (KMO) coefficient were applied to verify the method applicability’... The KMO is relevant in the context of FA, but not PCA, which uses the "Kaiser rule".

We apologize for the mistake, we actually performed FA and this was duly correct into the text; line 132.

2. lines 128-129. 'KMO≥0.6 was considered appropriate in the partial correlation between diet variables’. Values of 0.6 are considered not best choices, would be at least >0.7 to be considerable as intermediate values according to Kaiser (1975)

 We acknowledge the reviewer for the comment and agree that KMO>0.7 would be more appropriate. We highlight that our study was based on secondary data and the diet was assessed considering only 22 healthy eating markers. Dietary studies using more detailed tools, as 24 hour recalls or food frequency questionnaires, have found similar KMO. Please find some references below:

Previdelli ÁN et al. Using Two Different Approaches to Assess Dietary Patterns: Hypothesis-Driven and Data-Driven Analysis. Nutrients. 2016 Oct;8(10):593. 

Santos IKS dos, Conde WL. Trend in dietary patterns among adults from Brazilian state capitals. Rev Bras Epidemiol. 2020;23:e200035.

Carvalho CA, et al. Methods of a posteriori identification of food patterns in Brazilian children: a systematic review. Ciênc. saúde coletiva [Internet]. 2016 Jan [citado 2021 Jan 19] ; 21( 1 ): 143-154. Disponível em: http://www.scielo.br/scielo.php?script=sci_arttext&pid=S1413-81232016000100143&lng=pt. http://dx.doi.org/10.1590/1413-81232015211.18962014.

Liu X et al. Dietary patterns and the risk of esophageal squamous cell carcinoma: A population-based case–control study in a rural population. Clin Nutr. 2017 Feb 1;36(1):260–6.

3. line 180. ‘The highest p-value for the Cronbach Alpha test was 0.55’ Cronbach Alpha test is lower than references recommendations, indicates as limit cut above 0.70 as acceptable, however ≥0.80 would better. (Griethuijsen et al., 2014, Cortina, J. M., 1993).

We agree that Cronbach’s Alpha test was lower than the recommended values by the above references. We assumed 0.55 as bordering. However, factor loads were higher than 0.3 for each food while the total KMO=0.61 and the total variance was 0.402. According to similar results in the literature, we concluded that factor analysis was applicable to the data.

4. line 180-181. ‘Milk was not included in any of the dietary patterns due to a low factorial load’. I believe that information must be explored in more depth.

In the final factor analysis, only food with factor load higher than 0.3 were maintained, thus milk was not included in any dietary pattern.

5. line 230. 'which explained 40% of the diet variability'. Is this percentage considered enough for the total variance explanation?

Based on the previous studies we considered 40% of the data variability to be enough. Please find below some examples of studies in nutritional epidemiology, which had similar results: 

Schulze, MB, et al. Dietary patterns and their association with food and nutrient intake in the European Prospective Investigation into Cancer and Nutrition (EPIC)–Potsdam study. British Journal of Nutrition, v. 85, n. 3, p. 363–373, mar. 2001

Cunha DB, et al. Association of dietary patterns with BMI and waist circumference in a low-income neighbourhood in Brazil. Br J Nutr. 2010 Sep;104(6):908–13.

Nascimento S, et al. Dietary availability patterns of the Brazilian macro-regions. Nutr J. 2011 Jul 28;10:79.

Previdelli ÁN, et al. Using Two Different Approaches to Assess Dietary Patterns: Hypothesis-Driven and Data-Driven Analysis. Nutrients. 2016 Oct;8(10):593. 

Liu X, et al. Dietary patterns and the risk of esophageal squamous cell carcinoma: A population-based case–control study in a rural population. Clin Nutr. 2017 Feb 1;36(1):260–6.

6. line 265. ‘Several factors could explain healthier eating habits in older individuals.’ There is only one reference to explain the whole background. The authors should rewrite with more information about these hypotheses.

We appreciate the reviewer’s comment. We revised the sentence and provided a more suitable scientific background to support our hypotheses. 

Reviewer #2: This study examined the association between sociodemographic characteristics and the main Brazilian dietary patterns. This manuscript addresses an interesting nutrition topic using data of the Brazilian population. The study has its strengths and, overall, the article is well written, but I have some major comments.

1. Title: Aren't age and sex sociodemographic characteristics? The use of the term "sociodemographic characteristics" is not enough in the title?

We acknowledge the reviewer’s comment. Based on that, we rewrote the title more clearly. We highlight that the variables related to health and lifestyle characteristics were incorporated into the manuscript as suggested therefore, we changed the title accordingly. 

2. Title: “Brazilian dietary pattern” or “Brazilian dietary patterns”?

We apologize for the mistake and agree that the correct term is dietary patterns.

3. Introduction: There are many sentences in the first paragraph of "Introduction" text without references. This needs to be revised. Example (Page 2-line 47): “Changes in dietary patterns associated with government initiatives have contributed to reducing disease prevalence related to nutritional deficiencies.”

We apologize for the mistake. The paragraph was revised and the references were added.

4. Design and Study Population: Is this a cross-sectional study? Why is this not described in the study design?

We acknowledge reviewer’s comment and duly added the study’s design description.

5. Variables (Page 4-line 93): “The selected variables were age, sex, colour/race, marital status, education, area of residence, macro-region, socioeconomic status, physical activity, tobacco use, alcohol intake, self-related health status and multimorbidity.” This section of the manuscript describes behavioral, health and multimorbidity variables. However, only sociodemographic variables were explored in the study. What is the main reason for adopting this procedure? Exploring behavioral, health and multimorbidity characteristics with dietary patterns would make the article more robust and interesting.

We acknowledge the thoughtful comment. We agree that exploring variables related to health and lifestyle characteristics make the manuscript more interesting. These variables were incorporated and discussed in the revised manuscript as independent variables. 

6. Statistical Analysis: The principal component analysis (PCA) appears to have been well conducted. The food consumption questionnaire was composed of 22 food items. Is this number of food items considered adequate? This could be highlighted or otherwise weighted among the study's limitations.

We agree that is a limitation of the study and we acknowledge it in the manuscript‘s limitations. We highlight, however, that we used secondary data from the largest health survey (n=60,202) conducted in Brazil, the Brazilian National Health Survey - BNHS. The BNHS addresses several aspects related to health, diet being one of them. Even without applying more detailed tools, such as 24 hour recalls (24hR) or Food Frequency Questionnaires (FFQ), the study was able to explain 40% of the diet variability. This percentage is higher than findings verified in many studies that used FFQ or 24hR. Please find below some examples of studies using other tools presenting similar results: 

Schulze, MB, et al. Dietary patterns and their association with food and nutrient intake in the European Prospective Investigation into Cancer and Nutrition (EPIC)–Potsdam study. British Journal of Nutrition, v. 85, n. 3, p. 363–373, mar. 2001

Cunha DB, et al. Association of dietary patterns with BMI and waist circumference in a low-income neighbourhood in Brazil. Br J Nutr. 2010 Sep;104(6):908–13.

Nascimento S, et al. Dietary availability patterns of the Brazilian macro-regions. Nutr J. 2011 Jul 28;10:79.

Liu X, et al. Dietary patterns and the risk of esophageal squamous cell carcinoma: A population-based case–control study in a rural population. Clin Nutr. 2017 Feb 1;36(1):260–6.

7. Statistical Analysis (Page 5-line 138): “Robust Poisson regression models allow correction for overestimation of associations when outcomes are not rare (>10% prevalence)(28).” I believe that there is a better reference for this information, considering cross-sectional studies. [Barros AJ, Hirakata VN. Alternatives for logistic regression in cross-sectional studies: an empirical comparison of models that directly estimate the prevalence ratio. BMC Med Res Methodol. 2003;3(1):21. DOI: 10.1186/1471-2288-3-21.]

We acknowledge the reviewer and inform you that the reference was cited in the manuscript. 

8. Statistical Analysis (Page 5-line 141): “Associations were presented as Incidence Rate Ratios (IRR) with 95% confidence intervals (95% CI).” Is this the adequate measure of effect for the data under analysis? Is there any reason for not using the Prevalence Ratio?

We acknowledge the thoughtful comment and apologize for the mistake. We agree that Prevalence Ratio is the appropriate measure since it is a cross-sectional study design. Based on that, we included the appropriated measure and interpretation to the results.

9. Statistical Analysis: The description of the multivariate analysis model is confusing and needs to be revised. For example: Have demographic variables been adjusted to each other? Were behavioral, health and morbidity variables considered only as possible confounders? Why these variables were not explored as independent variables? If these characteristics are not explored as independent variables, I believe that it would be more appropriate to remove them from the analysis of the present study.

We revised the topic and rewrite it. We hope it is clearer now. In the revised manuscript, the variables related to health and lifestyle characteristics were included and discussed as independent variables. We believe that the final version of the manuscript presents a more robust discussion about the association between dietary pattern and variables related to health and lifestyle.

10. Results (Table 1): The data interpretation using the estimated sample size becomes confused. Is this form of presentation necessary?

BNHS has a complex sample design. The sample design used conglomerates in three levels: primary units of sampling (first stage), household (second stage), and selection of the household adult resident (third stage). We considered this complexity in all analysis. For this reason, we believe that the interpretation using the estimated population size is more appropriate than mentioning only the sample size itself.

11. Results (Page 10-line 194): “Table 3 shows the associations between the first quartile (Q1 - reference quartile) and the last quartile (Q4) of each pattern.” This procedure was not described and justified in the Methods section of the manuscript. Why was not the first quartile (Q1) compared to the sum of the remaining three (Q2 + Q3 + Q4)?

We aimed to compare the two extremes of the diet (first and fourth quartiles). We believed that the sum of the Q2+Q3+Q4 would not capture the associations between these quartiles: people from the first and the second quartiles have similar scores for the dietary patterns, for example. Nevertheless, we included the tables describing associations between intermediate quartiles (Q1-Q2 and Q1/Q3) as Supporting Information material.

12. Results (Page 10-line 198): “In stratified analysis by macro-region, age showed a dose-response effect.” This sentence is not clear. Was this finding only for the regions and not for the national data (Brazil)?

We apologize and clarify that the dose-response effect was also observed in the national analysis. The correction was made in the manuscript; line 227. 

13. Results: Table 3 is too long. I believe that this should be reduced. As most of the findings are similar across regions, Table 3 should only include national results (Brazil) and possible differences between regions should be described only in the text of the results.

We appreciate the suggestion. In the revised manuscript, table 3 was reduced and present only the national findings; the main differences between regions are described in the text. The regional findings are now provided in the supporting information material. 

14. Results: Table 3. The comparison groups (categories) of dietary patterns should be indicated in the title or heading of the Table.

We acknowledge for the suggestion. We included in the tables’ captions.

15. Results: Table 3: The use of the following terms should be reviewed: “Crude” or Unadjusted? -- “(CI 95%)” or 95%CI? -- “<0.005” or <0.001?

We acknowledge the reviewer for point out these inconsistencies, which were rectified. In accordance with manuscripts published in this field, we choose the term "Crude" to refer to the Prevalence Ratios of the univariate analyses.

16. Results: Table 3 (footnote): What mean "mutually adjusted models"?

This footnote was revised and the term was deleted since we included health and lifestyle characteristics. We explored the association between dietary patterns - as the dependent variable - and variables related to health, lifestyle and sociodemographic characteristics. In the modeling, we have no independent variable as the main variable. All multivariate models were (mutually) adjusted for all significant independent variables, since these variables were not considered only as confounders.

17. Results: The ‘Results section’ should be thoroughly revised. Why the findings regarding ‘marital status’ and ‘skin color’ were not highlighted in the text, for example?

We revised the section and the findings related to marital status, skin color and those related to health and lifestyle characteristics were included in the revised manuscript.

18. Discussion: First paragraph of the Discussion: why the main findings for the associations between sociodemographic factors and dietary patterns were not pointed out in the text?

We acknowledge the reviewer to point out this issue. In the revised manuscript, the main findings were briefly described in the first paragraph as suggested.

19. Discussion (page 19-line 252): “The traditional Brazilian pattern is an intermediate pattern characterised by rice and beans consumption. This pattern was not identified in this study because the survey did not evaluate rice consumption.” Considering its regular consumption by the Brazilian population, what was the reason for this food item has not been evaluated?

This is a limitation of the study. BNHS prioritized the assessment of healthy eating food markers and rice was not included. We highlighted and discussed this gap in the revised manuscript. Beans with rice is the most traditional food combination in Brazil. 

20. Discussion (page 21-line 303): “Ethnicity is related to social inequalities and income distribution in Brazil and other developing countries.” This sentence needs reference (s).

We provided updated reference in the revised manuscript to support our sentence.

21. Conclusion: The conclusion (last paragraph of the discussion) could be more explored and expanded. This could contemplate a synthesis of all sociodemographic aspects that showed an association in the study, in addition to age.

We acknowledge for the suggestion. The conclusion was rewritten.

---

## [Editor Report · Decision Letter 1]

1 Feb 2021

Health, lifestyle and sociodemographic characteristics are associated with Brazilian dietary patterns: Brazilian National Health Survey

PONE-D-20-35214R1

Dear Dr. dos Santos,

We’re pleased to inform you that your manuscript has been judged scientifically suitable for publication and will be formally accepted for publication once it meets all outstanding technical requirements.

Kind regards,

Michele Drehmer, Ph.D

Academic Editor

PLOS ONE

Additional Editor Comments (optional):

I have no further comments.

---

## [Editor Report · Acceptance letter]

4 Feb 2021

PONE-D-20-35214R1 

Health, lifestyle and sociodemographic characteristics are associated with Brazilian dietary patterns: Brazilian National Health Survey 

Dear Dr. Monteiro dos Santos:

I'm pleased to inform you that your manuscript has been deemed suitable for publication in PLOS ONE. Congratulations! Your manuscript is now with our production department. 

Kind regards, 

on behalf of

Dr. Michele Drehmer 

Academic Editor

PLOS ONE